# Impact of Gender and Age on Claim Rates of Dread Disease and Cancer Insurance Policies in Taiwan

**DOI:** 10.3390/ijerph19010216

**Published:** 2021-12-25

**Authors:** Chu-Shiu Li, Chih-Jen Hung, Sheng-Chang Peng, Ya-Lee Ho

**Affiliations:** 1Department of Risk Management and Insurance, National Kaohsiung University of Science & Technology, Kaohsiung 824, Taiwan; chushiu.li@gmail.com; 2Ph.D. Program of Business, College of Business, Feng Chia University, Taichung 40724, Taiwan; 3Department of Anesthesiology, Taichung Veterans General Hospital, Taichung 40705, Taiwan; 4Department of Risk Management and Insurance, Ming Chuan University, Taipei 111, Taiwan; scpeng@mail.mcu.edu.tw; 5Department of Business Administration, Feng Chia University, Taichung 40724, Taiwan; yaleeho@gmail.com

**Keywords:** non-life insurance, dread disease insurance, cancer insurance, claim rate, Taiwan

## Abstract

In this paper, the impact of both gender and age on the claim rates of dread disease and cancer insurance policies were examined using unique data taken from Taiwan’s private health insurance policies issued by non-life insurers during the 2012 to 2015 policy years. Those aged 30–39 served as the reference group. For the total number of dread disease policies, male insureds had a higher non-cancer claim probability than female insureds, while an age under 20 was associated with much lower claim rates for dread disease policies than for ages over 50. The claim rate for dread disease policies increased rapidly beginning at age 40 for both cancerous and non-cancerous diseases amongst male insureds. Amongst female insureds, those under 20 had much lower claim rates for dread disease policies. Only those aged 50–59 had a higher claim rate for non-cancerous diseases. For the total number of cancer insurance policies, male insureds had lower claim rates than female insureds, with an upward trend being associated with age. For male (female) insureds aged over 40 (20), the claim rates of cancer increased with age.

## 1. Introduction

Cancer is an increasingly relevant global public health issue due to both aging populations and today’s prevailing unhealthy lifestyles. In Taiwan, according to annual data from the Ministry of Health and Welfare (MHW), cancer has been the leading cause of death among males and females since 1982. The number of new cancer cases was 116,131 in 2018, representing an increase of 4447 cases when compared with the number of new cancer cases in 2017. Cancer incidences in Taiwan have rapidly increased, including amongst younger people. Although the incidence of cancer in males has decreased slightly in recent years, the incidence of cancer in females continues to increase [1].

Female breast cancer and colorectal cancer were among the cancers that had the largest increases in case numbers in Taiwan in 2018. A possible explanation for this is that screening programs for these two cancers has led to their detection in the early stages (stage 0 and stage 1) [2]. The Taiwan Cancer Registry database has verified the effectiveness of nationwide cancer screening programs in helping to reduce the cancer burden in Taiwan. There was a 48.0% reduction in incidences of invasive cervical cancer from 1995 to 2006 in women who underwent triennial screening in Taiwan [3]. Among people receiving a fecal immunochemical test, there was a significant reduction (10%) in colorectal cancer mortality [4]. Women have long been informed that cancer is of particular concern and that they need to undergo periodic testing and physical checks [5]. A perceived cancer risk is a strong motivation for precautionary health behavior and is associated with participation in cancer screening programs [6,7]. Reductions in cancer mortality following the implementation of screening programs have been well documented [8,9]. In addition, there is the expectation that government-led promotions surrounding public health awareness, as well as convenient screening programs, result in increases in the number of people diagnosed with cancer. Decreasing mortality rates of cancer may increase the loss ratio for long-term cancer policies or indemnity health insurance policies.

In 2018, the Insurance Bureau of Taiwan’s Financial Supervisory Commission regulated dread disease insurance, which covers seven types of conditions and diseases, including acute myocardial infarction (AMI), coronary artery bypass graft, stroke, end-stage renal disease, cancer, paralysis and major organ transplantation. Six of the top 10 causes of death in Taiwan are covered by dread disease insurance [10]. Dread diseases have higher mortality rates and substantial out-of-pocket costs for treatment, causing both psychological stress and a higher financial burden for patients and their families. Therefore, owning affordable non-life health insurance policies such as cancer insurance or dread disease insurance provides some financial protection [11].

In Taiwan, the annual premiums for non-life cancer insurance policies are much lower than those for life cancer insurance policies, due to non-guaranteed renewal. For example, in 2012 the premium for a non-life (life) cancer policy for a 30-year-old male having one million NT$ of insured coverage was 617 NT$ (7000 NT$) for the first year. Thus, whether contracts include guaranteed renewal plays a key role in setting health insurance premiums. Although Taiwan’s National Health Insurance (NHI) provides free healthcare for patients with catastrophic diseases (including malignancies), certain targeted therapies for cancer management and some nutritional supplements for post-chemotherapy care are not covered. It has been estimated that the initial out-of-pocket medical expenses for cancer are as high as one million NT$, with the average expense being in the range of 100,000 to 700,000 NT$ [12]. The indirect costs of cancer, such as lost wages due to absenteeism during treatment in outpatient clinics or hospitalization, are also substantial. For most cancers, these indirect costs account for over 50% of the total economic burden in Taiwan [13]. A study on advanced gastric cancer in Taiwan estimated the indirect costs to be 342 million US$ at the national level or approximately 77% of the total costs [14]. Therefore, the increase in the demand for private health insurance (PHI) may be related to insufficient financial support from Taiwan’s NHI. According to statistics from the Taiwan Insurance Institute (TII), the annual revenue from one-year term premiums for non-life private individual health insurance policies increased steadily from 348 million NT$ in 2015 to 759 million NT$ in 2020, or approximately 2.2-fold [15]. For those in the general public, it is a common risk management strategy to own at least one type of PHI policy.

In Taiwan, average out-of-pocket medical expenditures per household per year increased from 17,726 NT$ after the implementation of NHI in 1996 to 57,000 NT$ in 2012 [16]. Small households with higher income levels (annual income greater than 450 thousand NT$) are more likely to have private and public insurance coverage [17]. In China, health and household financial statuses are significant determinants in the decision to buy PHI. Interestingly, individuals with a better health status are more likely to purchase PHI [18].

The aim of this study is to examine and compare the risk factors for claim probabilities for dread disease and cancer insurance policies. We focus on gender and age, which are two key factors for determining health insurance premiums and the purchasing of PHI [19]. We used the unique national individual-level data from Taiwan’s private non-life health insurance companies, for both dread disease insurance and cancer insurance, during the 2012 to 2015 policy years. Our empirical evidence provides valuable implications for health-related risk management and future premium setting by both the government and non-life insurance companies.

## 2. Materials and Methods

Data was derived from Taiwan’s national private dread illness insurance policies and cancer insurance policies issued by non-life insurers during the 2012–2015 policy years (or 2012–2016 calendar years), as well as from TII, a semi-governmental organization responsible for insurance industry data collection and governance. As required by law, all property and liability insurance companies in Taiwan are members of the Non-Life Insurance Association and must provide insurance policy information related to premium setting and claim records of the insured to TII.

In Taiwan, the term for a non-life insurance policy is one year, without a guarantee of renewal. Although we had access to data for four policy years, 2012–2015, data for each year were treated as new contracts. As TII data is encrypted, we could not observe whether contracts were either new or renewed, or if there was any switching of insurance companies or channel systems during each policy year.

The main purpose of this paper was to examine and compare the risk factors of gender and age for claim probabilities regarding dread disease and cancer non-life insurance policies. Logistic regression analysis was used to empirically assess the determinants of the claim rates for dread disease insurance and cancer insurance policies. Logistic regression results are presented in Tables 3 and 4 as adjusted odds ratios (ORs) and 95% confidence intervals (CIs). The claim rates for dread disease insurance policies and cancer insurance policies were also calculated. The dependent variable was binary and indicated whether a claim was filed for dread disease insurance or cancer insurance during a policy year. Independent variables for both types of health insurance policies included the characteristics of the insureds (gender and age), characteristics of the insurance policies (channel systems, insured amount, main contract, size of insurers, waiting period) and policy year. We focused on the analysis of the risk factors of gender and age. STATA/SE 16.0 software was used for data management and statistical analyses. The relationships between dependent variables and independent variables were verified by the *t* test, which is calculated using STATA software. The normality, variance heteroscedasticity and collinearity had each been checked prior to conducting the various tests. A *p*-value < 0.1 was considered statistically significant.

## 3. Results

### 3.1. Characteristics of the Sample

Table 1 presents the descriptive statistics of the samples. There were 90,901 dread disease insurance policies and 235,747 cancer insurance policies. On the whole, the claim rate for dread disease insurance (2 per thousand) was significantly lower than that for cancer insurance (3 per thousand). In addition, the average claim payment per dread disease insurance policy (627 NT$) was significantly lower than that per cancer insurance policy (1967 NT$).

In terms of demographic variables, 50% of dread disease policyholders were male and 50% were female. In addition, 43.6% of dread disease policyholders were aged under 20, 18.9% were aged 30–39, and 10% were aged over 50. A slightly higher percentage of males (52%) purchased cancer insurance than females (48%), and approximately 66% of cancer insurance policyholders were aged 30–59. Thus, cancer insurance policyholders were mainly middle-aged, while dread disease policyholders were mainly young.

There were three main channel systems for dread disease insurance policies: direct writer systems (34.9%), agent systems (34.8%) and broker systems (30.2%). For cancer insurance, policies were mainly purchased through agent systems (49%), followed by direct writer systems (30.7%) and broker systems (20.3%).

Regardless of the type of insurance, most policies were the main contract. This was particularly true for cancer insurance (98%). Approximately 90% of dread disease insurance policies had a waiting period of 30 days, while more than 85% of cancer insurance policies had a waiting period of 90 days. Most of the policies were underwritten by the top 3 insurers, which shows that the market concentration is high, especially for cancer insurance (up to 82%). To sum up, more types of illnesses were covered, premiums were lower, and waiting periods were shorter for dread disease insurance policies, when compared with cancer insurance policies.

Panel A of Table 2 shows the comparisons between claim policies and non-claim policies for dread disease insurance. Only 149 policies had claims, accounting for 0.16% of all policies, with average claim payments being approximately 382 thousand NT$ (about 566 thousand NT$ for the insured amount). There were no significant differences between insureds, with and without a claim, in terms of gender. However, the average age of insureds with a claim (47.7) was significantly higher than that of insureds without a claim (25.8). Policyholders with a dread disease insurance claim were mainly aged 40–59 (67.1%) or above. Amongst policyholders without a claim, 43.6% were in the under 20 age group. There was a significant difference in the age distribution between policyholders with and without a claim. Cancer was the main illness type for dread disease insurance claims, accounting for 71.1% of the total. Circulatory system diseases (such as AMI, other cardiovascular diseases, and stroke) accounted for 10.7%, with others (such as kidney disease, other causes) accounting for 18.1%.

Panel B of Table 2 shows the comparisons between cancer insurance policies with and without claim. There were 687 policies with claim, accounting for 0.29% of all policies, with the average claim amount being approximately 675 thousand NT$ (about 847 thousand NT$ for the insured amount). There were no significant differences in terms of gender, while the average age of insureds with a claim was significantly higher than that of insureds without a claim. The proportions of insureds with a claim in the 40–49, 50–59, and over 60 age groups were 18.8%, 38.0%, and 37.3%, respectively, accounting for 94.1% of the total, while the proportions of insureds without a claim in the same age groups only accounted for 57.6% of the total. This difference was significant. The proportion of policyholders with a cancer insurance claim that were over age 50 was larger than that of policyholders with a dread disease insurance claim.

It is worth mentioning that for non-life health insurance policies with a claim, purchases were mainly made through the agent channel system, followed by the direct writer system. These two channels accounted for 75% of dread disease insurance policy purchases, and up to 93% of cancer insurance policy purchases. In general, claim policies were positively correlated with channel market shares, particularly for cancer insurance policies.

Figure 1 shows the claim rates of different age groups for both dread disease insurance and cancer insurance policies for total insureds (A), male insureds (B), and female insureds (C). For both types of health insurance, the claim rates increased gradually with age, especially for those aged over 40 (an increase of 2.7–2.9‰ for dread disease insurance and more than 3‰ for cancer insurance). The claim rates of dread disease insurance were higher than those of cancer insurance in each age group, particularly among those aged 40–49 and 50–59 with a difference of more than 1‰.

When we examined each subgroup by gender, the pattern of claim rates for male insureds was similar to that of total insureds. There were significant differences in claim rates between dread disease insurance policies and cancer insurance policies for those aged over 40. In contrast, amongst female insureds, claim rates for dread disease insurance policies were higher than those of cancer insurance policies in the groups aged younger than 50. Those in the groups aged over 50 had opposing results, which indicates that women are at higher risk of cancer than of dread diseases after the age of 50. Claim rates for both types of health insurance increased with age. However, there were dissimilar patterns for gender. For dread disease insurance, male insureds had a higher risk of claim than female insureds, after the age of 40. For cancer insurance, female insureds in the 30–59 age group had higher risk of claim than male insureds (Figure 1).

### 3.2. Regression Results 

We ran adjusted OR regression for the characteristics of the insureds (gender and age), characteristics of the insurance policies (channel systems, insured amounts, main contract or not, size of insurers and waiting periods), and also the policy years for both dread disease insurance (Table 3) and cancer insurance (Table 4). We only present the coefficients of gender and age, which are the main risk factors used in health insurance premium settings. Other independent variables were treated as control variables. We also show the complete logistic regression results in the Appendix A.

Table 3 shows the adjusted ORs from logistic regressions for dread disease insurance. Panel A presents the empirical results related to dread disease insurance claims. Table 1 indicates that cancer accounts for approximately 70% of dread disease insurance claims. Thus, the adjusted OR regression results of cancer (Panel B) and non-cancer illness (Panel C) related dread disease insurance claims are presented separately.

In Table 3, Panel A, there is no significant correlation between the claim rate for dread disease insurance and gender. However, there is a positive and significant correlation with age. Compared with the insureds in the 30–39 age group, those in the 40–49 age group had a four-times higher claim rate (OR, 5.445; 95% CI, 2.959–10.017), while those in the 50–59 age group and over 60 age group had an eight-times higher (OR, 9.633; 95% CI, 5.157–17.995) and 12-times higher claim rate (OR, 13.729; 95% CI, 6.726–28.024), respectively. This indicates that the risk of dread diseases gradually increases with age.

Amongst males, compared with the 30–39 age group, there were no significant differences in claim rates for dread disease insurance in the younger age groups, while adjusted ORs increased rapidly in the 40–49 age group (OR, 38.628; 95% CI, 5.255–283.917), 50–59 age group (OR, 71.509; 95% CI, 9.648–529.999), and over 60 age group (OR, 92.720; 95% CI, 11.867–724.414). In contrast, amongst females, compared with the 30–39 age group, those in the under 20 age group had significantly lower claim rates (OR, 0.086; 95% CI, 0.024–0.308). Female insureds in the over 40 age groups had a significantly higher probability of incurring claim, which is consistent with male insureds. However, the increases in adjusted ORs were much smaller.

Table 3, Panel B shows the empirical results of claims related to cancer for dread disease insurance policies. As cancer accounted for approximately 70% of claims for dread disease policies, we further examined whether the estimation results differed from those of Panel A. In the total sample, there were no differences in the effects of gender on cancer claims, which is consistent with the results in Panel A. However, in the under 20 age group, cancer claim rates were reduced by 86% (OR, 0.142; 95% CI, 0.045–0.453). The rates were around 3.5 times higher in the 40–49 age group than in the 30–39 age group (OR, 4.456; 95% CI, 2.265–8.770), which differed from the results in Panel A. On sub-group analysis, Panel B results were consistent with Panel A results for both male and female insureds, while the incidence of cancer claims increased at a rate of 19 to 60 times after the age of 40 among male insureds. 

Table 3, Panel C presents the claim incidences for non-cancer dread diseases for the total sample, male sample and female sample. For the total sample, male insureds were more likely to file a claim for non-cancer dread disease than female insureds (OR, 2.212; 95% CI, 1.165–4.199). When compared with the 30–39 age group, the younger age groups showed no significant differences in claims for non-cancer dread diseases, while the adjusted ORs of claim rates increased in older age groups. In the 40–49 age group, 50–59 age group, and over 60 age group, the rates were 9-times higher (OR, 10.534; 95% CI, 2.415–45.949), 22-times higher (OR, 23.18; 95% CI, 5.271–101.947), and 19-times higher (OR, 20.198; 95% CI, 3.287–124.116), respectively.

It is worth noting that the claim probability of non-cancer dread diseases increased after the age of 40. From the perspective of gender, the claim incidence of non-cancer dread diseases showed a significant increase only in the 50–59 age group amongst female insureds, with significant increases occurring after the age of 40 amongst male insureds.

Table 4 shows the adjusted OR regression results for policyholders with cancer insurance. For the total sample, gender had a significant impact on claims related to cancer, with a 28% lower claim probability for male insureds when compared with female insureds (OR, 0.716; 95% CI, 0.615–0.833). With increasing age, the risk of cancer gradually and significantly increases. Compared with insureds in the 30–39 age group, those in the under 20 and 20–29 age groups had 84% lower (OR, 0.160; 95% CI, 0.049–0.522) and 75% lower (OR, 0.251; 95% CI, 0.089–0.705) claim rates, respectively.

There were significant increases in the risk of incurring a claim related to cancer amongst middle-aged insureds with cancer insurance. Compared with the insureds in the 30–39 age group, the insureds in the 40–49 age group, 50–59 age group, and over 60 age group had approximately 2 times (OR, 2.927; 95% CI, 2.013–4.256), 6.5 times (OR, 7.547; 95% CI, 5.293–10.762), and 12 times (OR, 12.936; 95% CI, 9.031–18.53) a higher risk of filing a claim due to cancer, respectively.

On sub-group analysis, Table 4 shows that the claim incidence for cancer amongst male insureds over 40 significantly increases, particularly in the 50–59 age group (OR, 14.950; 95% CI, 6.94–32.205) and over 60 age group (OR, 35.795; 95% CI, 16.623–77.078). Compared with the 30–39 age group, the likelihood of filing a claim due to cancer was 92% less in the 20–29 age group (OR, 0.079; 95% CI, 0.011–0.579) amongst female insureds. In contrast, female insureds in the 40–49 age group, 50–59 age group, and over 60 age group were 1.5 times (OR, 2.505; 95% CI, 1.63–3.849), 4.8 times (OR, 5.805; 95% CI, 3.868–8.712), and 6.6 times (OR, 7.596; 95% CI, 4.98–11.587) more likely to file a claim due to cancer, respectively. 

Finally, we report the effects of other control variables. In the Appendix A, the logistic regression of claim occurrence in total dread disease insurance policies, for the controlled variables of channel, both channel systems of broker and direct response had significantly higher claim occurrence than direct writer system. In addition, policies with a waiting period of 90 days had lower claim occurrence than those with 60 days. On sub-group analysis of dread disease insurance policies, compared with direct writer system, broker channel system had significantly higher claim occurrence for the male insureds while direct response channel system had significantly higher claim occurrence for the female insureds. Moreover, male insureds holding dread disease insurance policies with a waiting period of 60 days had significantly higher claim occurrence than those with other two types of waiting periods.

The Appendix A shows the logistic regression of claim with cancer in dread disease insurance policies. The big-three insurers had significantly less claim occurrence in the groups of total insureds and male insureds. Regarding the channel systems, compared with the direct writer system the direct response system in groups of total insureds and female insureds had significantly higher occurrence of claim with cancer (OR 16.342 and 28.309, respectively). Only male insureds with waiting period of 60 days had a significantly higher occurrence of claiming cancer than others.

Appendix A reports the logistic regression of claim with non-cancer in dread disease insurance policies. The broker channel system had higher occurrence of claim with non-cancer than direct writer system in groups of total insureds and male insureds. The higher insurance coverage, groups of total insureds and male insureds had lower claim rate with non-cancer.

Appendix A indicates the logistic regression of claim occurrence in cancer insurance policies. The big-three insurers had significantly less claim occurrence in total insureds group and sub-groups. In addition, the broker channel system had a lower occurrence of claim than the direct writer system in groups of total insureds and female insureds (OR 0.686 and 0.581, respectively).

## 4. Discussion

In this study, the impacts of gender and age on claim probability were compared between dread disease insurance policies and cancer insurance policies by controlling for other variables. Amongst all dread disease claims, those related to cancer accounted for 71.1% of the total. Those involving circulatory system diseases were highest among non-cancer diseases and accounted for approximately 11% of claims. Based on the total sample, there were no significant effects regarding gender on the claim rates of dread diseases, although male insureds had higher claim rates than female insureds for non-cancer related diseases. These findings were consistent with the results of a systematic review which showed that male stroke incidence rates are 33% higher than those for females [20]. For the total dread disease claim data, we found that male insureds received lower claim payments than female insureds, which indicates more severe dread diseases occurring amongst females with a claim than amongst males with a claim, after controlling for other variables. Our empirical results are consistent with research performed surrounding circulatory system diseases. For example, stroke had a greater impact on females than males, with stroke-related outcomes being poorer in females [21,22,23].

Compared with the 30–39 age group, those in the 20–29 age group displayed significantly lower claim rates for dread disease insurance. Claim rates began to increase after the age of 50 in the total sample. When we further investigated the types of diseases for which claims were filed, there were no differences in the claim rates for dread diseases among the younger age groups, while the claim rates increased rapidly amongst males over the age of 40 for both cancer and non-cancer related diseases. In contrast, amongst female insureds, those aged under 20 had significantly lower claim rates for dread disease insurance. Only the 50–59 age group had higher claim rates for non-cancer related diseases. Chen et al. investigated the epidemiology and disease burden of ischemic stroke in Taiwan (from 2000 to 2005) and found that stroke mortality increased from 50 to 2300 per 100,000 individuals in the 50–90 age group, with a higher prevalence amongst females [24]. Lee et al. found that the age- and gender-adjusted incidence of acute AMI remained relatively stable from 49.8 per 100,000 population in 2009 to 50.7 in 2015 [25]. However, the incidence of AMI increased 30.3% and 29.4% in young male and female populations (<55 years), respectively. This incidence either decreased or remained unchanged in other age groups.

For total cancer insurance policies, we found that male insureds had lower claim rates than female insureds, and that claim rates gradually increased with age. In terms of gender having an effect, our evidence is inconsistent with the results of a recent study performed by Huang and Chen which showed that within the same birth cohort, the cancer incidence of males is higher and increases faster than that of females [1]. The incidence of cancer in males has decreased slightly, although the incidence of cancer in females has continued to increase in Taiwan in recent years. The different results for the total samples in Taiwan may be due to a difference in databases. Huang and Chen used data taken from the Taiwan Cancer Registry System for the years 1988 to 2016 [1]. In the present study, we investigated non-life cancer insurance policies for the years 2012 to 2016, with a limitation of insurable age set at under 60. In addition, since Taiwan has been promoting various cancer screening programs since 2004 [3,26], many cancers have been diagnosed at the early stage, such as breast cancer at stage 0, for which a dread disease insurance claim may not be filed.

On sub-group analysis of cancer insurance policies, the incidence of cancer amongst male insureds over 40 years of age significantly increased, particularly in the 50–59 and over 60 age groups. Compared with the female insureds in the 30–39 age group, the claim rate of cancer was significantly lower for the 20–29 age group and higher among other age groups. A meta-analytic review concluded that individuals perceiving themselves to be at a greater risk of developing breast cancer were more likely to undertake cancer screening [27]. Similarly, under the country’s universal health care plan, a Korean survey found that perceived cancer risk possibility was significantly associated with having private cancer insurance only amongst females, and that females with private cancer insurance were more likely to follow precautionary healthy behavior than males [28]. The perceived cancer risk within the Korean population was also associated with their participation in cancer screening [29].

It should be noted that there were several limitations to this study. Firstly, we used individual non-life health insurance policies, and focused on the comparisons of the impacts of gender and age on claim probabilities regarding dread disease and cancer insurance policies. Therefore, insured individuals over 60 years of age accounted for only 2.6% of dread disease insurance policyholders and 11.8% of cancer insurance policyholders, which was inconsistent with the natural age distribution of the population. Our empirical results are based upon policyholders with one-year non-life health insurance contracts. International data has also shown that young consumers are the main insured group for major diseases [30]. Secondly, we could only divide diseases into three categories, although seven types of dread diseases are covered by dread disease insurance. Thirdly, we could not observe whether the policy information was for renewed contracts, or if there was any switching of insurance companies during each policy year. Fourthly, most of our logistic regressions had a low R-squared value, however, the LR chi2 tests were significant, indicating that independent variables of logistic models still had effective explanatory power. Finally, we used the individual health policies during the 2012–2016 calendar years rather than recent year data.

## 5. Conclusions

This study contributes to the understanding of the risk factors surrounding gender and age for claim rates regarding dread disease and cancer insurance policies. To the best of our knowledge, this is the first empirical study to have investigated age and gender as risk factors for dread disease insurance claims based upon national non-life health insurance policies in Taiwan. From the results, gender and age groups had different claim probabilities and claim severities for cancer and non-cancer diseases. The incidence of non-cancer dread diseases significantly increased only in the 50–59 age group for female insureds and in the age groups over 40 for male insureds. For male (female) insureds aged over 40 (20), the claim rates for cancer increased with age. Identifying the role of gender and age in claims amongst the private health insureds within a universal health care system may improve outcomes by providing healthcare professionals with additional insight regarding gender dependent and age-specific tailored approaches to treatment. Our empirical evidence provides policy implications for insurers when setting premiums, as well as for government institutions when formulating health risk management strategies. Further research surrounding the impact of private health insurance in the universal health care system is still needed. In addition to our analysis of the unique and entire national private dread illness insurance policies and cancer insurance policies during the 2012–2016 calendar years, future researchers may find it worthwhile to investigate whether the effects of gender and age on the claim rates of dread disease and cancer insurance policies may have differed during the recent years.

## Figures and Tables

**Figure 1 ijerph-19-00216-f001:**
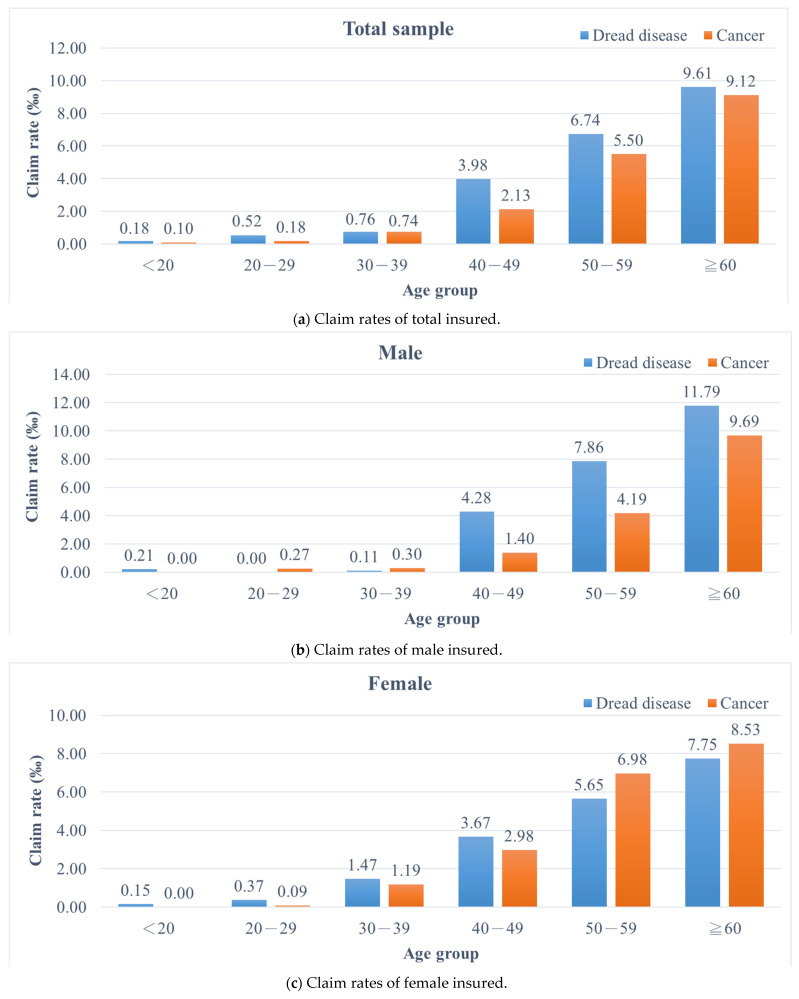
Claim rates of different age groups for dread disease and cancer insurance policies by gender (Claim rate = # of claims for each age group/# of insurance policies for each age group). (**a**) Claim rates of total insureds; (**b**) claim rates of male insureds; (**c**) claim rates of female insureds.

**Table 1 ijerph-19-00216-t001:** Descriptive statistics of dread disease insurance and cancer insurance variables.

	(1) Dread Disease Insurance (N_D_ = 90,901)		(2) Cancer Insurance (N_C_ = 235,747)		(3) *p*-Value (1) vs. (2)
	Mean	SD	Mean	SD	
Claim rate (%)	0.002	0.040	0.003	0.054	<0.001 ***^b^
Average claim costs (NT$)	626.553	20,775.294	1967.167	45,489.194	<0.001 ***^a^
Male	0.494	0.500	0.520	0.500	<0.001 ***^a^
Age (mean)	25.834	17.393	40.633	15.952	
Age					<0.001 ***^b^
<20	0.436	0.496	0.129	0.336	
20–29	0.126	0.332	0.093	0.291	
30–39	0.189	0.391	0.200	0.400	
40–49	0.147	0.354	0.257	0.437	
50–59	0.077	0.266	0.202	0.401	
≥60	0.026	0.160	0.118	0.323	
Main contract	0.740	0.438	0.980	0.139	<0.001 ***^b^
Big three insurers	0.663	0.473	0.821	0.383	<0.001 ***^b^
Distribution channel					<0.001 ***^b^
Direct writer system	0.349	0.477	0.307	0.461	
Agent system	0.348	0.476	0.490	0.500	
Broker system	0.302	0.459	0.203	0.402	
Direct response system	0.001	0.028	0.000	0.014	
Premium	1847.217	3785.144	5609.457	8392.151	<0.001 ***^a^
Insured amount (NT$)	610,974.027	378,175.740	1,148,999.186	759,483.732	<0.001 ***^a^
Waiting period					<0.001 ***^a^
30 days	0.892	0.311	0.131	0.338	
60 days	0.089	0.285	0.016	0.127	
90 days	0.019	0.137	0.852	0.355	
Year					<0.001 ***^a^
2012	0.261	0.439	0.181	0.385	
2013	0.260	0.439	0.222	0.416	
2014	0.249	0.432	0.277	0.447	
2015	0.230	0.421	0.320	0.467	

Note: ^a^
*t*-test; ^b^ Chi-square test; *** *p* < 0.01.

**Table 2 ijerph-19-00216-t002:** Descriptive statistics of claim vs. non-claim policy variables.

	(1) Claim Policy		(2) Non-Claim Policy		(3) *p*-Value
	Mean	SD	Mean	SD	(1) vs. (2)
Panel A: Dread disease insurance (N = 90,901)					
Average claim costs (NT$)	382,243.470	343,854.274			
Male	0.510	0.502	0.494	0.500	0.695 ^b^
Age (mean)	47.738	12.927	25.798	17.377	<0.001 ***^,b^
Age					<0.001 ***^,b^
<20	0.047	0.212	0.436	0.496	
20–29	0.040	0.197	0.126	0.332	
30–39	0.087	0.283	0.189	0.391	
40–49	0.356	0.480	0.146	0.353	
50–59	0.315	0.466	0.076	0.266	
≥60	0.154	0.363	0.026	0.159	
Illness type ^c^					
Cancer	0.711	0.455			
Circulatory system	0.107	0.311			
Others	0.181	0.386			
Distribution channel					0.017 **^,b^
Direct writer system	0.342	0.476	0.349	0.477	
Agent system	0.409	0.493	0.348	0.476	
Broker system	0.242	0.430	0.302	0.459	
Direct response system	0.007	0.082	0.001	0.028	
Premium	4757.544	5490.674	1842.438	3779.913	<0.001 ***^,a^
Insured amount (NT$)	565,771.812	39,4261.508	611,048.241	37,8146.589	0.163 ^a^
N	149		90,752		
Panel B: Cancer insurance (N = 235,747)					
Average claim costs (NT$)	675,041.920	506,057.174			
Male	0.493	0.500	0.520	0.500	0.167 ^b^
Age (mean)	55.237	9.710	40.590	15.947	<0.001 ***^,a^
Age					<0.001 ***^,b^
<20	0.003	0.054	0.130	0.336	
20–29	0.006	0.076	0.093	0.291	
30–39	0.051	0.220	0.200	0.400	
40–49	0.188	0.391	0.258	0.437	
50–59	0.380	0.486	0.201	0.401	
≥60	0.373	0.484	0.117	0.322	
Distribution channel					
Direct writer system	0.330	0.471	0.307	0.461	<0.001 ***^,b^
Agent system	0.595	0.491	0.489	0.500	
Broker system	0.074	0.262	0.203	0.402	
Direct response system	0.000	0.000	0.000	0.014	
Premium	10,598.603	11,377.412	5594.876	8377.547	<0.001 ***^,a^
Insured amount (NT$)	846,746.143	713,117.862	1,149,242.518	759,434.562	<0.001 ***^,a^
N	687		235,060		

Note: ^a^
*t*-test. ^b^ Chi-square test. ^c^ In addition to cancers, dread diseases include those of the circulatory system (such as AMI, other cardiovascular diseases, brain, stroke and hemorrhage), along with others such as kidney disease and other causes. ** *p* < 0.05, *** *p* < 0.01.

**Table 3 ijerph-19-00216-t003:** Adjusted odds ratios (ORs) of logistic regressions for dread disease insurance.

	Total (N_DT_ = 90,901)		Male (N_DM_ = 44,861)		Female (N_DF_ = 45,992)	
	Adjusted OR	95% CI	Adjusted OR	95% CI	Adjusted OR	95% CI
Panel A: Dependent variable = Claim occurrence						
Male	1.089	[0.788, 1.505]				
Age (Ref = 30–39)						
<20	0.188	[0.074, 0.476]	1.414	[0.156, 12.801]	0.086 ***	[0.024, 0.308]
20–29	0.639	[0.242, 1.686]	3.058	[0.277, 33.812]	0.424	[0.136, 1.324]
40–49	5.445 ***	[2.959, 10.017]	38.628 ***	[5.255, 283.917]	2.637 ***	[1.308, 5.316]
50–59	9.633 ***	[5.157, 17.995]	71.509 ***	[9.648, 529.999]	4.336 ***	[2.085, 9.019]
≥60	13.729 ***	[6.726, 28.024]	92.720 ***	[11.867, 724.414]	7.577 ***	[3.125, 18.372]
Pseudo *R*^2^	0.1209		0.1671		0.1080	
LR chi2	267.15		187.45		117.43	
P > chi2	0.0000		0.0000		0.0000	
Log likelihood	−970.929		−467.136		−484.766	
Panel B: Dependent variable = Cancer claim occurrence						
Male	0.823	[0.56,1.211]				
Age (Ref = 30–39)						
<20	0.142 ***	[0.045,0.453]	0.938	[0.084,10.496]	0.070 **	[0.015,0.325]
20–29	0.372	[0.103,1.339]	2.994	[0.270,33.167]	0.130 *	[0.017,1.024]
40–49	4.456 ***	[2.265,8.770]	19.012 ***	[2.494,144.919]	2.943 ***	[1.389,6.235]
50–59	7.334 ***	[3.634,14.802]	43.969 ***	[5.765,335.344]	3.522 ***	[1.537,8.069]
≥60	12.103 ***	[5.550,26.394]	60.033 ***	[7.484,481.562]	7.844 ***	[3.032,20.293]
Pseudo *R*^2^	0.1273		0.1867		0.1209	
LR chi2	209.33		137.94		109.25	
P > chi2	0.0000		0.0000		0.0000	
Log likelihood	−717.207		−300.481		−397.197	
Panel C: Dependent variable = Non-cancer claim occurrence						
Male	2.212 **	[1.165, 4.199]				
Age (Ref = 30–39)						
<20	0.436	[0.072, 2.658]	0.296	[0.041, 2.155]	0.160	[0.014, 1.802]
20–29	2.174	[0.361, 13.08]			1.743	[0.287, 10.594]
40–49	10.534 ***	[2.415, 45.949]	10.26 ***	[2.342, 44.938]	0.751	[0.067, 8.42]
50–59	23.18 ***	[5.271, 101.947]	15.651 ***	[3.390, 72.261]	8.89 ***	[1.724, 45.843]
≥60	20.198 ***	[3.287, 124.116]			5.420	[0.440, 66.841]
Pseudo *R*^2^	0.1436		0.1591		0.1310	
LR chi2	106.89		75.83		33.29	
P > chi2	0.0000		0.0000		0.0026	
Log likelihood	−318.734		−200.369		−110.424	

Note: (1) Regressions were controlled by other variables, including Constant, Distribution channel, Insured amount, Main contract, Big three insurers, Waiting period and Policy year. (2) In Panels A and B, Stata deleted the variable of the Direct response system due to collinearity when running the regression of male data, thus 48 male observations were not included. (3) In Panel C of total data, Stata deleted the variable of the Direct response system due to collinearity when running the regression, thus 70 observations were not included. (4) In Panel C of male data, Stata deleted the variables of age 20–29, age ≥ 60, Direct response system and Waiting period of 90 days due to collinearity when running the regression, thus 5397 male observations were not included. (5) In Panel C of female data, Stata deleted the variables of the Direct response system and Waiting period of 90 days due to collinearity when running the regression, thus 937 female observations were not included. (6) After the treatment of collinearity, the variance inflation factor (VIF) for all the independent variables in all models was less than 10. Adjusted OR refers to odds ratio; 95% CI refers to 95% confidence intervals. * *p* < 0.10, ** *p* < 0.05, *** *p* < 0.01.

**Table 4 ijerph-19-00216-t004:** Adjusted odds ratios (ORs) of logistic regressions for cancer insurance.

	Total (N_CT_ = 235,699)		Male (N_CM_ = 122,465)		Female (N_CF_ = 98,304)	
	Adjusted OR	95% CI	Adjusted OR	95% CI	Adjusted OR	95% CI
Male	0.716 ***	[0.615, 0.833]				
Age (Ref = 30–39)						
<20	0.160 ***	[0.049, 0.522]	0.705	[0.18,2.755]		
20–29	0.251 ***	[0.089, 0.705]	0.924	[0.239,3.579]	0.079 **	[0.011,0.579]
40–49	2.927 ***	[2.013, 4.256]	4.838 ***	[2.183,10.724]	2.505 ***	[1.63,3.849]
50–59	7.547 ***	[5.293, 10.762]	14.586 ***	[6.775,31.4]	5.805 ***	[3.868,8.712]
≥60	12.936 ***	[9.031, 18.53]	34.229 ***	[15.926,73.569]	7.596 ***	[4.98,11.587]
Pseudo R^2^	0.079		0.0931		0.0538	
LR chi2	742.47		395.12		270.12	
P > chi2	0.0000		0.0000		0.0000	
Log likelihood	−4325.4522		−1923.6253		−2377.5382	

Note: (1) Regressions were controlled by other variables, including Constant, Distribution channel, Insured amount, Main contract, Big three insurers, Waiting period and Policy year. (2) In the total (male) data, for collinearity, Stata deleted the variables of the Direct response system and waiting period, thus 48 (24) observations were not included. (3) In the female, for collinearity, Stata deleted the variables of age <20, Direct response system and Waiting period due to collinearity, thus 14,954 female observations were not included. (4) After the treatment of collinearity, the variance inflation factor (VIF) for all the independent variables in all models was less than 10. Adjusted OR refers to odds ratio; 95% CI refers to 95% confidence intervals. ** *p* < 0.05, *** *p* < 0.01.

## Data Availability

Data supporting the reported results can be found in TII.

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
