# Peer review of "Impact of Gender and Age on Claim Rates of Dread Disease and Cancer Insurance Policies in Taiwan"

_ijerph, 2021, doi:10.3390/ijerph19010216_

Round 1

Reviewer 1 Report

I think this paper deals with a very important topic in the policy making of  risk management field.

However, there are some points that are not fully explained before reaching the conclusion, which I will discuss below.

Since the reasons for focusing on gender and age are tenuous, you will need to cite supporting articles and provide evidence that the factors of gender and age are statistically significant.

The reliability of the test results is not clear because normality and other checks have not been performed before conducting the various tests. Has the normality and other statistical features been checked?

In the sections of the paper where Tables 3 and 4 are cited and explained in the paper, the ratio values are difficult to understand from the tables.
(ex. four-fold higher claim rate and eight-fold higher claim rate @ P8, 9-fold higher, 22-fold higher and 19-fold higher @ P9, 2-fold, 6.5-fold and 12-fold @ P9)

In particular, the explanation of the M-shaped pattern on page 10 is not clear, even though it is one of the conclusions of this paper, as to which part of the table the M-shaped pattern can be seen.

Adding an explanation of the peculiarities of Taiwan's insurance policy, or adding a description of insurance policies for similar diseases in other countries to the discussion or conclusion may lead to the possibility of developing a methodology for obtaining similar data in other countries.

When a table crosses pages, it would be better to have the field names appear on the next page as well.

Author Response

Response to Reviewer 1 Comments

Point 1: Since the reasons for focusing on gender and age are tenuous, you will need to cite supporting articles and provide evidence that the factors of gender and age are statistically significant. 

Response 1: Thank you so much for your comments. Based on your comments and suggestions, we add a reference (#19) to support our focus on gender and age. Please see the last paragraph in page 2 of the revised manuscript.

Point 2: The reliability of the test results is not clear because normality and other checks have not been performed before conducting the various tests. Has the normality and other statistical features been checked?

Response 2: Thank you so much for your comments and suggestions. Based on your comments and suggestion, we have checked collinearity effect through VIF for all logistic and OLS regression models. We also add the description in the final paragraph of the section of Materials and Methods (please see page 3 of revision manuscript).

For our all logistic models, after considering the treatments of collinearity, the VIFs of all independent variables are lower than 10 (please see Appendix Tables 1 and 2 at the end of this responding report). We also add the notes statements related to all information which variables were excluded and then how many observations were dropped in each model at the bottoms of Tables 3 and 4. Please see Tables 3 and 4 in pages 8 and 9 of our revision manuscript, respectively.

For our all logistic models, after considering the treatments of collinearity, the VIFs of all independent variables are lower than 10 (please see Appendix Tables 1 and 2 at the end of this responding report). We also add the notes statements related to all information which variables were excluded and then how many observations were dropped in each model at the bottoms of Tables 3 and 4. Please see Tables 3 and 4 in pages 8 and 9 of our revision manuscript, respectively.

For our OLS regression model, we used the Shapiro-Wilk W test and White’s test to check normality and heteroskedasticity for OLS regression based on your comments. However, our results showed that the p-value is very small for all tests, indicating that we would have to reject that residuals are normally distributed. The null hypothesis of White’s test is that the variance of the residuals is homogenous. The p-value is very small except for male and female samples of dread disease insurance and the female sample of cancer insurance, we would have to reject the hypothesis and accept the alternative hypothesis that the variance is not homogenous (please see Appendix Table 3 at the end of this responding report). Therefore, we decide that we will not report the OLS regression results in pages 10 and 11 of this revision manuscript for concerning the conditions of normality and heteroscedasticity for OLS regression models. We also delete the part concerning OLS regression in the final paragraph of the section of Materials and Methods (please see the strikethrough in page 3 of revision manuscript).

Point 3: In the sections of the paper where Tables 3 and 4 are cited and explained in the paper, the ratio values are difficult to understand from the tables. (ex. four-fold higher claim rate and eight-fold higher claim rate @ P8, 9-fold higher, 22-fold higher and 19-fold higher @ P9, 2-fold, 6.5-fold and 12-fold @ P9)

Response 3: We really appreciate your comments. Based on your comments, we have revised “-fold” to “times” in pages 10 and 11 of revision manuscript.

Point 4: In particular, the explanation of the M-shaped pattern on page 10 is not clear, even though it is one of the conclusions of this paper, as to which part of the table the M-shaped pattern can be seen.

Response 4: Thank you so much for your comments and suggestions. As age is a set of variables, including <20, 20-29, 30-39, 40-49, 50-59 and ≥60, the insureds aged 30-39 were used as the reference group when estimating the coefficients of age variables for female insureds. The concerning of collinearity among independent variables based on your comments above, Stata deleted the variable of age <20 when running the regression. Then the new complete odds ratios of all age variables were 0.079, 1, 2.5, 5.8 and 7.7 without an M-shaped pattern. We have revised the description for female insureds. Please see the last second paragraphs of pages 10 and 12, and the sections of Conclusion and Abstract.

Point 5: Adding an explanation of the peculiarities of Taiwan's insurance policy, or adding a description of insurance policies for similar diseases in other countries to the discussion or conclusion may lead to the possibility of developing a methodology for obtaining similar data in other countries.

Response 5: Thank you so much for your comments. Based on your comments, in addition to the previous description of Taiwan’s private health insurance (PHI) policy (please see the third paragraph of page 2), we have added some related empirical evidences from other countries with PHI in the section of Discussion (please see the last 3 sentences in page 12). We also have revised the section of Conclusion (please see page 13 of revision manuscript).

Point 6: When a table crosses pages, it would be better to have the field names appear on the next page as well.

Response 6: Thank you so much for your comments and suggestions. We have re-edited our Tables accordingly.

(Appendix Tables 1-3 are presented in the next pages.) 

Appendix Table 1. VIF of independent variables for logistic regressions

VIF

Variable

Total sample

Male sample

Female sample

Panel A: Dread disease insurance policies

Male

1.00

Age (Ref=30-39)

<20

2.04

2.00

2.10

20-29

1.49

1.43

1.55

40-49

1.53

1.50

1.56

50-59

1.32

1.30

1.34

≥60

1.15

1.13

1.18

Big-three insurers

1.87

1.87

1.89

Channel (Ref=Direct writer system)

Agent system

2.07

2.07

2.07

Broker system

1.83

1.79

1.87

Direct response system

1.01

1.01

1.00

Waiting period (Ref=60 days)

30 days

2.00

1.95

2.06

90 days

1.51

1.49

1.53

Ln(Insured amount)

1.58

1.55

1.61

Policy year (ref. 2012)

2013

1.48

1.49

1.48

2014

1.48

1.49

1.47

2015

1.47

1.49

1.46

Main contract

2.11

2.14

2.10

Panel B: Cancer insurance policies

Male

1.00

Age (Ref=30-39)

<20

1.59

1.62

1.55

20-29

1.34

1.35

1.33

40-49

1.71

1.77

1.66

50-59

1.64

1.68

1.61

≥60

1.48

1.49

1.48

Big-three insurers

1.71

1.74

1.68

Channel (Ref=Direct writer system)

Agent system

2.57

2.67

2.47

Broker system

1.89

1.91

1.86

Direct response system

1.00

1.00

1.00

Ln(Insured amount)

1.52

1.52

1.53

Policy year (ref. 2012)

2013

1.74

1.73

1.75

2014

1.87

1.85

1.88

2015

1.97

1.95

1.99

Main contract

1.26

1.23

1.29

Appendix Table 2. VIF of independent variables for OLS regressions

VIF

Variable

Total sample

Male sample

Female sample

Panel A: Dread disease insurance

Male

1.14

Age (Ref=30-39)

<20

1.09

1.18

1.06

20-29

1.13

1.27

3.62

40-49

3.91

4.45

4.68

50-59

5.45

6.81

4.33

≥60

5.44

7.27

6.59

Big-three insurers

5.52

4.95

3.19

Channel (Ref=Direct writer system)

Agent system

2.46

2.14

Broker system

1.60

1.80

1.08

Direct response system

Waiting period (Ref=60 days)

30 days

4.74

6.47

4.22

90 days

8.20

9.61

7.88

Ln(Insured amount)

1.59

1.72

1.59

Policy year (ref. 2012)

2013

1.76

1.73

1.85

2014

1.88

1.84

1.99

2015

1.91

1.84

2.12

Main contract

1.20

1.27

1.19

Appendix Table 3. Normality test and heteroscedasticity test for OLS regressions

Shapiro-Wilk W test

White’s test

Obs

W

Prob>z

chi2

Prob>chi2

Panel A: Dread disease insurance

Total sample

149

0.9153

0.0000

130.3400

0.0094

Male sample

76

0.9423

0.0018

42.6100

0.2080

Female sample

73

0.8438

0.0000

70.8800

0.1814

Panel B: Cancer insurance

Total sample

687

0.4072

0.0000

184.4500

0.0000

Male sample

339

0.4666

0.0000

170.2900

0.0000

Female sample

348

0.4020

0.0000

54.9800

0.7820

Reviewer 2 Report

The article looked at an area of interest, by evaluating the risk factors for claim probabilities for dread disease and cancer insurance policies, with a major focus on the effect of gender and age. However, I have the following concerns/comments;

I was wondering why the analysed was narrowed to 2012 to 2015. Is there no additional recent data? This need to be justified considering we are in 2021 now.

The method needs to be improved to increase the replicability of the study. For instance, how many people were ensured within this time? Did you enrol all people within the time? Any inclusion/exclusion criteria? Etc

Also, your estimation of rates, relationships (t test, chi-square) need to be made clear in the analysis section of the method. Additionally, you need to specify the model you adopted for the regression (e.g. forward, backward stepwise, etc) in the method section.

It appears you only conducted binary logistic regression, with no control of any variable. While this is good in providing an interpretation of the individual result, I think there is a need for multivariate logistic regression (using Adjusted Odds Ratio) in addition to the binary. This will account for the effect of one variable while controlling the other. E.g., what is the effect of age if you control for gender? and vice-versa. Additionally, you may as well consider controlling for the effect of the other independent variables (channel systems, insured amount, main contract, size of insurers, waiting period, and policy year) in addition to the age or gender. You simply need to put them in the multivariate logistic regression model. That way, your current results will remain, but you will have an additional interpretation of the findings to make a better meaning. Otherwise, those additional independent variables have less meaning in the study.

I further found it difficult to follow up with the M-shaped pattern for the association between claim rate for cancer and age among female insureds. If I got it right, you were referring to table 4, of which the OR were 0.079, 2.5, 5.8, 7.7 and 0.52. Perhaps it’s a different shape rather than M. If you are referring to a different result, then you need to make it clear.

You further need to acknowledge the low R-squared in the limitation, which indicates the strength of the model. Furthermore, the use of data from the year 2012 to 2015 also needs to be acknowledged as the finding may not be the case in the current situation.

Regards.

Author Response

Response to Reviewer 2 Comments

Point 1: I was wondering why the analyzed was narrowed to 2012 to 2015. Is there no additional recent data? This need to be justified considering we are in 2021 now. 

Response 1: Thank you so much for your comments. This paper uses the unique entire national private dread illness insurance policies and cancer insurance policies issued by ten non-life insurers during the 2012-2015 “policy years” from the Taiwan Insurance Institute (TII). Since we analyzed policy year data, so we examined whether polices claimed or not until December 30, 2016. For examine if the insured purchase the non-life cancer insurance policy in December 15, 2015, then we need to examine this insurance policy until December 14, 2016.

Yes, unfortunately we had no more recent data, this is our data limitation. In fact, we got these data for research in 2017 and we spent more than one year to organize and analyze the relevant empirical results in detail. Therefore we submit our paper until 2021. We have added this limitation in the last sentence of the section of Discussion and advocate executing further update researches in the section of Conclusion.

Point 2: The method needs to be improved to increase the replicability of the study. For instance, how many people were ensured within this time? Did you enroll all people within the time? Any inclusion/exclusion criteria?

Response 2: Thank you so much for your comments. Based on your comments and suggestions, we have added the notes related to how many numbers of observations were excluded due to the collinearity problem when we used the Stata software. Please see the bottoms of Tables 3 and 4 in pages 8 and 9 of our revision manuscript, respectively.

For our all logistic models, after considering the treatments of collinearity, the VIFs of all independent variables are lower than 10 (please see Appendix Tables 1 and 2 in the end of this responding report). We also add the notes statements related to all information which variables were excluded and then how many observations were dropped in each model at the bottoms of Tables 3 and 4. Please see Tables 3 and 4 in pages 8 and 9 of our revision manuscript, respectively.

Point 3: Also, your estimation of rates, relationships (t test, chi-square) need to be made clear in the analysis section of the method. Additionally, you need to specify the model you adopted for the regression (e.g. forward, backward stepwise, etc) in the method section.

Response 3: Thank you so much for your comments. The relationship between a dependent variable and an independent variable is verified by t test, which is calculated by STATA software. We have added the description in the last third sentence of the section of Materials and Methods (please see page 3 of revision manuscript).

In addition, in our paper the estimation models of exploring the important influential factors on claim rates consider all characteristics of insureds (age and gender) and insurance policies (channel systems, insured amount, main contract, size of insurers, waiting period and policy year) instead of using the stepwise regression method. We also have updated the description in the last second sentence of the section of Materials and Methods (please see page 3 of revision manuscript).

Point 4: It appears you only conducted binary logistic regression, with no control of any variable. While this is good in providing an interpretation of the individual result, I think there is a need for multivariate logistic regression (using Adjusted Odds Ratio) in addition to the binary. This will account for the effect of one variable while controlling the other. E.g., what is the effect of age if you control for gender? and vice-versa. Additionally, you may as well consider controlling for the effect of the other independent variables (channel systems, insured amount, main contract, size of insurers, waiting period, and policy year) in addition to the age or gender. You simply need to put them in the multivariate logistic regression model. That way, your current results will remain, but you will have an additional interpretation of the findings to make a better meaning. Otherwise, those additional independent variables have less meaning in the study.

Response 4: Thank you so much for your comments. All results of logistic regression models were estimated based on controlling other independent variables (channel systems, insured amount, types of contract, size of insurers, waiting period and policy year) in addition to the age or gender. Based on your comments and suggestions, we have added the related description in the last second sentence of the section of Materials and Methods (please see page 3 of the revised manuscript). We also present the complete logistic regression results in the Supplement based on your comments (please see the end of the revised manuscript in pages 16-19).

Point 5: I further found it difficult to follow up with the M-shaped pattern for the association between claim rate for cancer and age among female insureds. If I got it right, you were referring to table 4, of which the OR were 0.079, 2.5, 5.8, 7.7 and 0.52. Perhaps it’s a different shape rather than M. If you are referring to a different result, then you need to make it clear.

Response 5: Thank you so much for your comments. As age were a set of variables, including <20, 20-29, 30-39, 40-49, 50-59 and ≥60, the insureds aged 30-39 were used as the reference group when estimating the coefficients of age variables for female insureds. The concerning of collinearity among independent variables based on the other reviewer’s comments, Stata deleted the variable of age <20 when running the regression. In the current version, the new complete odds ratios of all age variables were 0.079, 1, 2.5, 5.8 and 7.7 without an M-shaped pattern. We have revised the description for female insureds. Please see the last second paragraph of page 10 and the second sentence of the last paragraph in page 12, the middle part of the section of Conclusion and the final part of the section of Abstract.

Point 6: You further need to acknowledge the low R-squared in the limitation, which indicates the strength of the model. Furthermore, the use of data from the year 2012 to 2015 also needs to be acknowledged as the finding may not be the case in the current situation.

Response 6: Thank you so much for your comments. Based on your comments and suggestions, we have discussed these limitations in the last 2 sentences of the section of Discussion and in the final part of the section of Conclusion (please see page 13 of the revised manuscript).

(Appendix Tables 1-3 are presented in the next pages.) 

Appendix Table 1. VIF of independent variables for logistic regressions

VIF

Variable

Total sample

Male sample

Female sample

Panel A: Dread disease insurance policies

Male

1.00

Age (Ref=30-39)

<20

2.04

2.00

2.10

20-29

1.49

1.43

1.55

40-49

1.53

1.50

1.56

50-59

1.32

1.30

1.34

≥60

1.15

1.13

1.18

Big-three insurers

1.87

1.87

1.89

Channel (Ref=Direct writer system)

Agent system

2.07

2.07

2.07

Broker system

1.83

1.79

1.87

Direct response system

1.01

1.01

1.00

Waiting period (Ref=60 days)

30 days

2.00

1.95

2.06

90 days

1.51

1.49

1.53

Ln(Insured amount)

1.58

1.55

1.61

Policy year (ref. 2012)

2013

1.48

1.49

1.48

2014

1.48

1.49

1.47

2015

1.47

1.49

1.46

Main contract

2.11

2.14

2.10

Panel B: Cancer insurance policies

Male

1.00

Age (Ref=30-39)

<20

1.59

1.62

1.55

20-29

1.34

1.35

1.33

40-49

1.71

1.77

1.66

50-59

1.64

1.68

1.61

≥60

1.48

1.49

1.48

Big-three insurers

1.71

1.74

1.68

Channel (Ref=Direct writer system)

Agent system

2.57

2.67

2.47

Broker system

1.89

1.91

1.86

Direct response system

1.00

1.00

1.00

Ln(Insured amount)

1.52

1.52

1.53

Policy year (ref. 2012)

2013

1.74

1.73

1.75

2014

1.87

1.85

1.88

2015

1.97

1.95

1.99

Main contract

1.26

1.23

1.29

Appendix Table 2. VIF of independent variables for OLS regressions

VIF

Variable

Total sample

Male sample

Female sample

Panel A: Dread disease insurance

Male

1.14

Age (Ref=30-39)

<20

1.09

1.18

1.06

20-29

1.13

1.27

3.62

40-49

3.91

4.45

4.68

50-59

5.45

6.81

4.33

≥60

5.44

7.27

6.59

Big-three insurers

5.52

4.95

3.19

Channel (Ref=Direct writer system)

Agent system

2.46

2.14

Broker system

1.60

1.80

1.08

Direct response system

Waiting period (Ref=60 days)

30 days

4.74

6.47

4.22

90 days

8.20

9.61

7.88

Ln(Insured amount)

1.59

1.72

1.59

Policy year (ref. 2012)

2013

1.76

1.73

1.85

2014

1.88

1.84

1.99

2015

1.91

1.84

2.12

Main contract

1.20

1.27

1.19

Appendix Table 3. Normality test and heteroscedasticity test for OLS regressions

Shapiro-Wilk W test

White’s test

Obs

W

Prob>z

chi2

Prob>chi2

Panel A: Dread disease insurance

Total sample

149

0.9153

0.0000

130.3400

0.0094

Male sample

76

0.9423

0.0018

42.6100

0.2080

Female sample

73

0.8438

0.0000

70.8800

0.1814

Panel B: Cancer insurance

Total sample

687

0.4072

0.0000

184.4500

0.0000

Male sample

339

0.4666

0.0000

170.2900

0.0000

Female sample

348

0.4020

0.0000

54.9800

0.7820

Round 2

Reviewer 2 Report

Thank you for given me the chance to re-review the manuscript.

The manuscript is much improved and most of my concerns are addressed.

It is great that you controlled for the other variables in your main analysis (LR) and you have made that clear in the revised manuscript.

My only minor observations now are;

1) It is more appropriate to label your regression as 'Adjusted OR' rather than just OR since the models have controlled for the other variables. 

2) Consider stating (perhaps in a short paragraph) at the end of the LR result description (sub-section 3.2) about the key controlled variables that were significant since table 3 and 4 only show the variables of interest (age and gender). For instance, in the result of supplemental table 4; of the controlled (or adjusted) variables, the 'broker system' channel is the only significant variable affecting overall cancer insurance claim rate for different age groups/male-female gender (Supplemental table 4).

The above will give more meaning as to why controlling for certain variables is important. You may narrow this final interpretation to the total claim rates.

Great effort.

Regards

Author Response

Point 1: It is more appropriate to label your regression as 'Adjusted OR' rather than just OR since the models have controlled for the other variables.  

Response 1: Thank you so much for your comments. Using the Track Changes function of MS Word, we have revised the “OR” with “adjusted OR” in our manuscript (ijerph-1488686) based on your suggestion.

Point 2: Consider stating (perhaps in a short paragraph) at the end of the LR result description (sub-section 3.2) about the key controlled variables that were significant since table 3 and 4 only show the variables of interest (age and gender). For instance, in the result of supplemental table 4; of the controlled (or adjusted) variables, the 'broker system' channel is the only significant variable affecting overall cancer insurance claim rate for different age groups/male-female gender (Supplemental table 4).

Response 2: Thank you so much for your comments. Based on your comments and suggestions, we have added a paragraph to describe the findings of each Supplement Table at the end of the section of Results (please see the final paragraph of page 10 and the first three paragraphs of page 11).